# Using the Bayesian Network to Map Large-Scale Cropping Intensity by Fusing Multi-Source Data

**Jianbin Tao [1], Wenbin Wu [2,*] and Meng Xu [1]**

[1] Key Laboratory for Geographical Process Analysis & Simulation of Hubei province/School of Urban and Environmental Sciences, Central China Normal University, Wuhan 430079, China; taojb@mail.ccnu.edu.cn (J.T.); xumeng1216@mails.ccnu.edu.cn (M.X.)

[2] Key Laboratory of Agricultural Remote Sensing, Ministry of Agriculture/Institute of Agricultural Resources and Regional Planning, Chinese Academy of Agricultural Sciences, Beijing 100081, China

* Correspondence: wuwenbin@caas.cn

**Abstract:** Global food demand will increase over the next few decades, and sustainable agricultural intensification on current cropland may be a preferred option to meet this demand. Mapping cropping intensity with remote sensing data is of great importance for agricultural production, food security, and agricultural sustainability in the context of global climate change. However, there are some challenges in large-scale cropping intensity mapping. First, existing indicators are too coarse, and fine indicators for measuring cropping intensity are lacking. Second, the regional, intra-class variations detected in time-series remote sensing data across vast areas represent environment-related clusters for each cropping intensity level. However, few existing studies have taken into account the intra-class variations caused by varied crop patterns, crop phenology, and geographical differentiation. In this research, we first presented a new definition, a normalized cropping intensity index (CII), to quantify cropping intensity precisely. We then proposed a Bayesian network model fusing prior knowledge (BNPK) to address the issue of intra-class variations when mapping CII over large areas. This method can fuse regional differentiation factors as prior knowledge into the model to reduce the uncertainty. Experiments on five sample areas covering the main grain-producing areas of mainland China proved the effectiveness of the model. Our research proposes the framework of obtain a CII map with both a finer spatial resolution and a fine temporal resolution at a national scale.

**Keywords:** cropping intensity index; regional differentiation; Bayesian network; prior knowledge; MODIS time-series

## 1. Introduction

Agricultural production is the foundation of human survival and development, and cropland is the main resource for agricultural production and the source of human civilization. Because urbanization develops rapidly, a large amount of cropland has been converted into construction lands [1–3]. At the same time, as the population continues to grow, the structure of the diet changes and non-food consumption of agricultural products increases, the demand for agricultural products are growing all the time [4,5]. How to alleviate this human-land conflict and improve agricultural land use intensity (or cropping intensity) on existing cropland is the focus of the international community. In China, due to the shortage in the agricultural labor force caused by rapid urbanization and the impact of the market economy on the farmers' income, the cropping frequency in some regions has shown a downward trend. Abandonment or idleness of cropland in winter is very common. Therefore, the cropping intensity and its spatio-temporal pattern have evoked widespread concern in academia and industry.

One of the major challenges in cropping intensity mapping is that definitions for fine cropping intensity measurements are lacking [6]. Most of the existing studies have defined cropping intensity as cropping frequency (single cropping or double cropping, etc.) [4,7–15] or MCI (multiple crop indexes) [16–22]. Cropping frequency is a hard division of crop pattern that can lead to much information loss. MCI, though defined in diverse ways, is essentially an aggregation of cropping frequency on a spatial scale or a synthesis of cropping frequency on a time scale, or both. The agricultural statistics used in the calculation of MCI lack detailed spatial information [8,23], and ignore the spatial heterogeneity of cropping intensity within the administrative districts. Some studies have also defined the cropping intensity from the aspect of input and (or) output, production management, and technology progress [24,25]. In general, these definitions are also measurement based on statistical data. Overall, the existing definitions cannot meet the requirements of fine cropping intensity mapping, and high spatial and temporal resolution cropping intensity datasets are lacking.

The second challenge is the problem of intra-class variations that are often ignored in large-scale cropping intensity mapping. Dominated by regional differentiation and crop phenology, the phenomena of SODS (same object with different spectrums) and DOSS (different objects with same spectrum) are very common in large-scale vegetation mapping [26]. Wardlow argued that vegetation index profiles were affected by regional variations in climate and management practices, which should be accounted for by setting-up individual profiles for each homogenous agro-region [27]. Gong proposed that improving mapping accuracy should focus on building more effective features, rather than optimizing the algorithm [28]. Chen believed that specialized knowledge derived from DEM, ecological zone, and other auxiliary data is an important way to overcome the difficulty of SODS and DOSS in remote sensing image classification [29]. However these within-class variations have seldom been fully addressed in the existing approaches [30]. Mapping cropping intensity over large areas is challenging, and the methods of cropping intensity mapping on a large-scale should be further strengthened.

The objective of this study was to develop methods to map cropping intensity on a national scale at a sub-pixel level. The work includes: (1) the development of a normalized index, the cropping intensity index (CII), to quantify cropping intensity precisely; (2) the development of a Bayesian network model fusing prior knowledge (BNPK) to fuse time-series MODIS (moderate resolution imaging spectroradiometer) data and regional differential information properly. The methods were calibrated and validated in five sample areas covering the main grain-producing areas of mainland China. The novelty of the research is that we presented a new indicator, CII, and a new method of estimating CII spanning vast geographical environments. The BNPK model can fuse prior knowledge about cropping intensity into the model, where the variations of vegetation index profiles over large areas can be accounted for.

## 2. Study Area and Data

### 2.1. The Study Area

Concentrated cropland areas in mainland China were chosen as the study areas. According to the distribution of croplands and its cropping intensity, climatic and topographic conditions, the whole country can be subdivided into 38 regions [31]. Five sample areas covered by complete Landsat scenes were selected, which span vast areas from the Songliao Plain, North China Plain, Middle-Upper Hanjiang River Valley, and the Middle-Lower Yangtze River Valley and have significant geographic gradients (Figure 1). For simplicity, we named these five samples areas as SP1, SP2, NCP, HRV, and YRV. Double cropping and single cropping coexist on the North China Plain, Middle-Upper Hanjiang River Valley, and the Middle-Lower Yangtze River Valley, while on the Songliao Plain, single cropping is the only crop pattern. There are no tri-season crops in the research area.

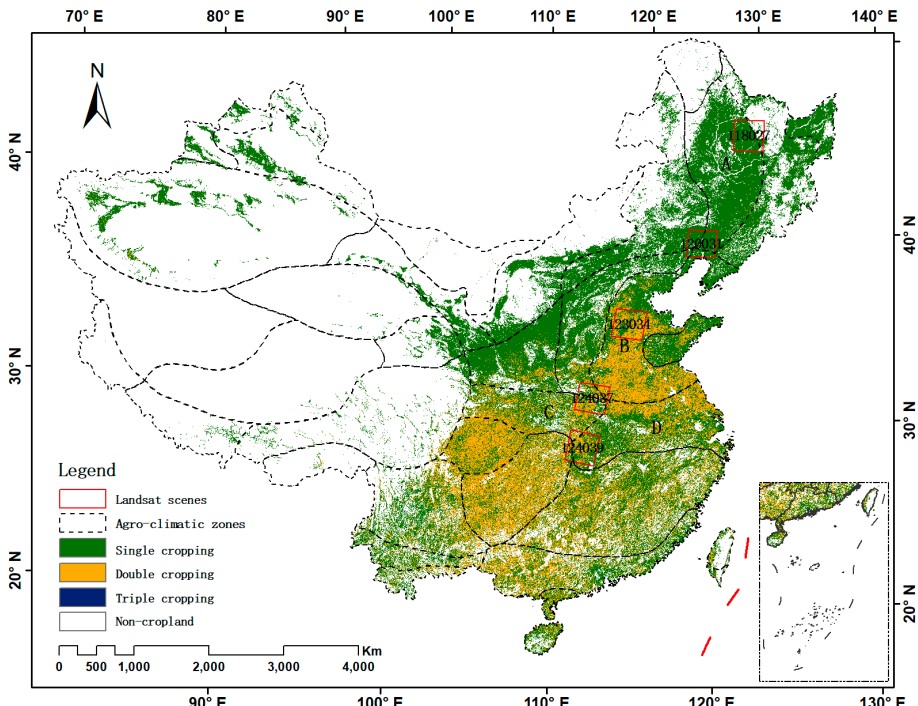

**Figure 1.** The research area, the sample areas (covered by five Landsat scenes), and the cropping frequency distribution in 2015 (Resource and Environment Data Cloud Platform, (http://www.resdc. cn/DOI, 2017. DOI:10.12078/2017122201). Regions A-D represent the Songliao Plain, North China Plain, Middle-Upper Hanjiang River Valley, and the Middle–Lower Yangtze River Valley, respectively.

## 2.2. The Data and Preprocessing

The data used in this study were the MOD13Q1 v006 data, consisting of MODIS products recorded by the EOS/Terra Satellite. The products include 250 m resolution NDVI (normalized difference vegetation index) and EVI (enhanced vegetation index) data, reflectance data, and quality control data, which were synthesized over 16 days based on the maximum value composite (MVC) method. EVI data layers were extracted as the vegetation indices datasets in this research. The EVI time-series was composed of 23 EVI composites covering a natural year. The products were corrected geometrically and atmospherically. The MODIS image sequence numbers of the tiles covering the research area were h26v04, h27v04, h27v05, h27v06, h28v05, and h28v06 (h: horizontal, v: vertical). In addition, a data pixel reliability layer was extracted from the MOD13Q1 v006 products, the spatial and temporal resolutions of which were consistent with the EVI dataset. MOD13Q1 was acquired from the National Aeronautics and Space Administration (NASA) website (http://modis-land.gsfc.nasa. gov) and covered 2015. Landsat 8 OLI images in 2015 with cloud cover less than 20% were obtained to generate the calibration and validation samples. The paths/rows of the Landsat scenes were 124–039, 124–037, 123–034, 120–031, and 118–027 (Figure 1 and Table 1). To facilitate the process of preparing the cropping frequency sample data through classification, the acquisition date was carefully selected to cover the phenology stage of the crops.

Geographical geometric correction, image clipping, and resampling were performed on the dataset. The method adopted for sampling was the nearest neighbor. The projection system was the Albers equal-area conic projection; the spheroid is the Clarke 1866 system, with a central meridian of 110°E and two standard parallels of 25°N and 47°N.

**Table 1.** Landsat 8 images used in the research.

| Paths/Rows | Acquisition Date | Cloud Cover (%) |
|:---:|:---:|:---:|
| 118/027 | 16/06/2015 | 1.76 |
| 120/031 | 10/03/2015 | 3.47 |
| | 26/03/2015 | 3.48 |
| | 13/05/2015 | 0.02 |
| 123/034 | 15/03/2015 | 9.02 |
| | 18/05/2015 | 0.26 |
| 124/037 | 17/01/2015 | 0.04 |
| | 09/05/2015 | 19.94 |
| 124/039 | 01/012015 | 12.51 |
| | 22/03/2015 | 19.19 |
| | 09/05/2015 | 6.69 |

## 3. Method

The proposed method for mapping cropping intensity from the time-series MODIS data consists of two components: (1) the definition of the cropping intensity index, and (2) the Bayesian network modeling and cropping intensity inference. Bayesian networks were used to model the nonlinear relationship between cropping intensity and multi-source data. The software packages used in our research for Bayesian network modeling and the subsequent corresponding model validation were Netica 5.02 and MATLAB.

### 3.1. Bayesian Network

The Bayesian network (BN) is a powerful mathematical model for reasoning about uncertainty, which combines probability theory and graph theory to express mutual relationships between variables. A Bayesian network is a DAG (directed acyclic graph) combined with a CPT (conditional probability table), where each node represents a random variable and the arcs linking the nodes represent the relationships between variables. The NB (naïve Bayesian network) is the simplest Bayesian network in terms of both structure and parameter learning and has been widely studied in many cases as a benchmark for comparison with new methods.

Assume that $X_1, X_2, \ldots, X_n$ are the random variables, $C$ is the class node, and $n$ is the number of random variables, then the following formula can be derived according to the Bayes formula and the chain rule:

$$P(C|X_1, X_2, \ldots, X_n) = \frac{P(X_1, X_2, \ldots, X_n|C)P(C)}{P(X_1, X_2, \ldots, X_n)} = \frac{P(X_1, X_2, \ldots, X_n, C)}{P(X_1, X_2, \ldots, X_n)} \tag{1}$$

The key step of calculating $P(C|X_1, X_2, \ldots, X_n)$ is to figure out the joint probability distribution on all nodes. In a NB, features are assumed to be mutually independent of each other within a given class, therefore a NB is fully defined by the conditional probabilities of each feature given the class. The class node is the parent of all feature nodes in a NB, and therefore the joint probability of all nodes can be written as follows according to the hypothesis of conditional independence between child nodes [32]:

$$P(X_1, X_2, \ldots, X_n, C) = P(C) \sum_{i=1}^{n} P(X_i|C) \tag{2}$$

According to Equations (1) and (2), we can obtain a simple way of calculating the posterior probability given the sample information and prior probability of the class node:

$$P(C|X_1, X_2, \ldots, X_n) = P(C) \prod_{i=1}^{n} \frac{P(X_i|C)}{P(X_i)} \tag{3}$$

where $P(C)$ and $P(X_i)$ are the prior probabilities of class node and features node, respectively. $P(X_i|C)$ is the conditional probability of feature $X_i$ given a class, usually obtained through training. $P(C|X_1, X_2, \ldots, X_n)$ is the posterior probability of the class node given the feature nodes.

### 3.2. Cropping Intensity Index

'Cropping intensity' can be defined in a number of ways and is usually defined as the cropping frequency. Cropping frequency was obtained by simply counting the peaks within the EVI time-series. This method can bring in bias because we do not know how to discriminate when it is not an obvious peak caused by mixed pixels and (or) diverse planting structure. Since the intensity and variability of the EVI time-series within a natural year can reflect the fragmentation of fields and areal coverage of a given crop pattern, we defined cropping intensity as sub-pixel level cropping frequency that is indirectly measured by the EVI time-series.

Here, we propose a normalized cropping intensity index (*CII*) to quantify cropping intensity:

$$CII = P(CI|EVI_1, EVI_2, \ldots, EVI_{23}) = P(CI) \prod_{i=1}^{23} \frac{P(EVI_i|CI)}{P(EVI_i)} \tag{4}$$

where *CI* is the cropping intensity from sample data. $EVI_1, EVI_2, \ldots, EVI_{23}$ is the MODIS EVI time-series. $P(CI)$ *and* $P(EVI_i)$ are prior information and are constants. $P(EVI_i|CI)$ is the conditional probability of $EVI_i$ given the cropping intensity, usually obtained through training. $P(CI|EVI_1, EVI_2, \ldots, EVI_{23})$ is the posterior probability of cropping intensity given the EVI time-series. The calculation of posterior probability is the same as Equation (3).

The calculation of the MODIS-like sample *CI* is based on the cropping frequency data from the Landsat classification result. $d_{crop}$ is the number of double cropping pixels within a MODIS pixel; $s_{crop}$ is the number of single cropping pixels (Figure 2); and n is the total number of Landsat pixels within a MODIS pixel. This equation gives a mean value of cropping frequency based on Landsat images within a MODIS pixel, with a data range from 0 to 1, indicating the 'intensity' of cropping.

$$CI = \frac{d_{crop} * 2 + s_{crop}}{2n} \tag{5}$$

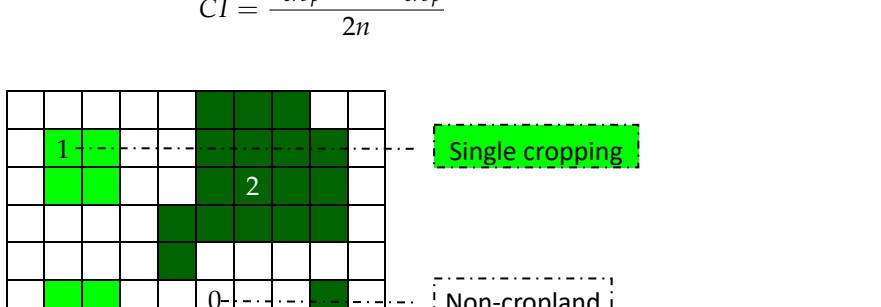

**Figure 2.** Schematic diagram for calculating sample *CI*.

It is assumed that there are pure double cropping pixels on the MODIS image, which is an ideal state. The *CI* of those pixels with homogeneous double cropping is close to 1; the *CI* of those pixels with homogeneous single cropping is approximately 0.5; the *CI* of non-cropland is 0.

The self-organizing data analysis technique (ISODATA), along with visual interpretation using Google Earth satellite images were utilized to classify the Landsat images. The classification maps were simply reclassified to a nominal single cropping/double cropping/non-cropland classification scheme to obtain the cropping frequency samples.

### 3.3. BNPK Model

We present the steps of building the BNPK model including fusing multi-source data using a Bayesian network (Figure 3), and the details of adding regional differentiation factor as prior knowledge into the model. As a spatial determinant of CII, zonal information was included in our model.

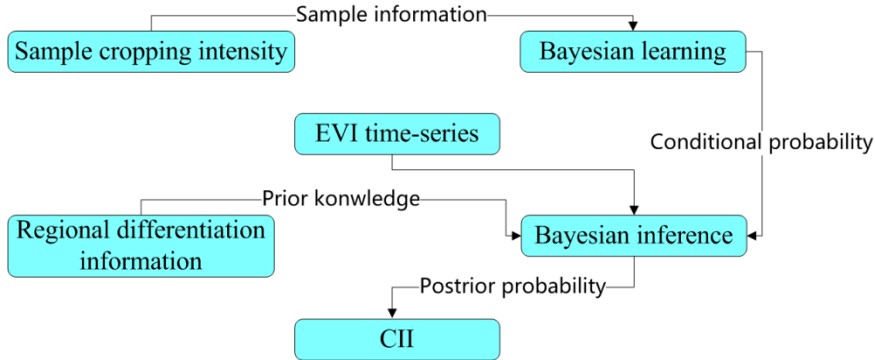

**Figure 3.** Flowchart of inferencing cropping intensity index.

### 3.3.1. Fusing Time-Series MODIS Data

When applying the Bayesian network to remote sensing applications, the parameters and the observed data are all regarded as random variables. We used the term variable to indicate any features. A Bayesian network was built to model the relationships between CII and time-series MODIS data, where the aforementioned EVI data were taken as independent variables and CII was taken as the dependent variable. Model construction included three stages: structure learning, parameter learning, and inference.

Since EVI profiles reflect the CII as a whole, we added the link manually to construct the DAG. The initial Bayesian network was composed of 23 child nodes defining the independent variables of interest (Figure 4).

All continuous variables must be converted to discrete quantities before the parameter learning because all probabilistic inference in Netica is done with discrete tables [30]. The Jenks natural breaks method [33] was used to discretize all the variables. The number of states for all child nodes was chosen based on a set of tests (10–30 states) to improve overall accuracy, and 13 states were finally selected. The CPT was estimated using the EM (expectation-maximization) algorithm that updates initial parameter estimation by iteratively refitting the data to the updated model until convergence. When the CPTs of each node have been defined (Table 2, using child node L1 as an example), the network is able to be 'solved' [33], and the CPTs and their changes can then easily be examined by each individual case. The BN provides a simple way to test a case, allowing the user to input evidence by assigning a value at a node. The effect of the case can then be examined by its assignment on other nodes through the propagation of probabilities. The rapid propagation of information through the network is one of the major advantages of the BN, which can be used to quickly view how observation at one node will affect the system as a whole [33].

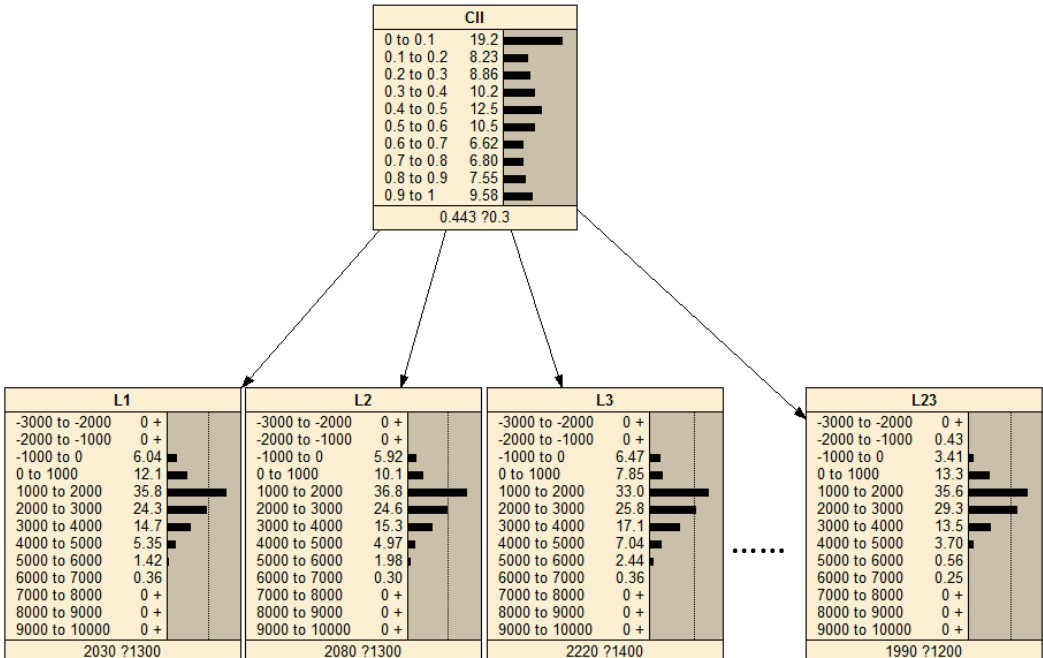

**Figure 4.** The initial Bayesian network model. L1, L2, etc., represent the enhanced vegetation index time-series.

In the parameter learning stage, all variables were set to observed values, and the model was built at this level (forward). In the inference stage, however, the CII was set to unobserved, and the results of the inference were then taken as the estimated CII values (backward). For a given case, the posterior probability distributions of CII were calculated according to Equation (4):

$$CII = P(CI|EVI_1, EVI_2, \ldots, EVI_{23}) = \prod_{i=1}^{23} CPT_i(j) \tag{6}$$

where $CPT_i$ is the CPT of child node $i$, $i \in (1, 2, \ldots, 23)$. $j$ denotes the sequence number of a column in $CPT_i$, $j \in (1, 2, \ldots, 13)$, and its value was determined according the EVI range of the corresponding child node.

CII values with maximum posterior probabilities were selected as the estimated values. The Bayesian network model was constructed and compiled and then run on the cases of the validation samples.

3.3.2. Adding Zone Information as Prior Knowledge

We introduced a zone node into the model, which represented the regional differentiation information. Here, the prior knowledge indicates the information about the CII available in addition to the training data from the MODIS data. The basis is that simply determining a model from a finite set of spatial proxies without prior knowledge is an ill-posed problem.

Unlike MODIS EVI data with interval measurements, zone data are a kind of data with nominal measurement, which cannot be modeled as a natural node in BN. We used codes 1, 2, 3, 4, and 5 to label the five sample areas in the Songliao Plain, North China Plain, Middle-Upper Hanjiang River Valley, and the Middle–Lower Yangtze River Valley respectively, so for each zone there is a separate CPT, which is locally trained. The zone node was added into the original BN model as the parent node of feature nodes. In this way, the zone information was fused into the initial model as prior knowledge to generate the final BNPK model (Figure 5).

**Table 2.** Conditional probability table of child node L1.

| CII \ EVI | −3000~−2000 | −2000~−1000 | −1000~0 | 0~1000 | 1000~2000 | 2000~3000 | 3000~4000 | 4000~5000 | 5000~6000 | 6000~7000 | 7000~8000 | 8000~9000 | 9000~10000 |
|---|---|---|---|---|---|---|---|---|---|---|---|---|---|
| 0–0.1 | 0.0 | 0.0 | 0.0 | 0.0 | 0.0 | 27.6 | 67.3 | 5.1 | 0.0 | 0.0 | 0.0 | 0.0 | 0.0 |
| 0.1–0.2 | 0.0 | 0.0 | 0.0 | 0.0 | 8.3 | 80.5 | 11.2 | 0.0 | 0.0 | 0.0 | 0.0 | 0.0 | 0.0 |
| 0.2–0.3 | 0.0 | 0.0 | 0.0 | 0.0 | 15.0 | 83.1 | 1.9 | 0.0 | 0.0 | 0.0 | 0.0 | 0.0 | 0.0 |
| 0.3–0.4 | 0.0 | 0.0 | 0.0 | 0.8 | 82.0 | 17.2 | 0.0 | 0.0 | 0.0 | 0.0 | 0.0 | 0.0 | 0.0 |
| 0.4–0.5 | 0.0 | 0.0 | 0.0 | 0.0 | 0.0 | 0.0 | 14.9 | 53.9 | 24.9 | 4.2 | 1.0 | 1.0 | 0.0 |
| 0.5–0.6 | 0.0 | 0.0 | 0.0 | 0.0 | 8.3 | 42.0 | 34.7 | 12.9 | 2.1 | 0.0 | 0.0 | 0.0 | 0.0 |
| 0.6–0.7 | 0.0 | 0.0 | 34.1 | 42.0 | 18.6 | 1.3 | 3.9 | 0.0 | 0.0 | 0.0 | 0.0 | 0.0 | 0.0 |
| 0.7–0.8 | 0.0 | 0.0 | 0.0 | 0.0 | 0.0 | 26.1 | 47.6 | 23.2 | 2.5 | 0.6 | 0.0 | 0.0 | 0.0 |
| 0.8–0.9 | 0.0 | 0.0 | 0.0 | 0.0 | 65.5 | 34.5 | 0.0 | 0.0 | 0.0 | 0.0 | 0.0 | 0.0 | 0.0 |
| 0.9–1.0 | 0.0 | 0.0 | 0.0 | 0.0 | 0.0 | 7.6 | 46.6 | 44.1 | 1.7 | 0.0 | 0.0 | 0.0 | 0.0 |

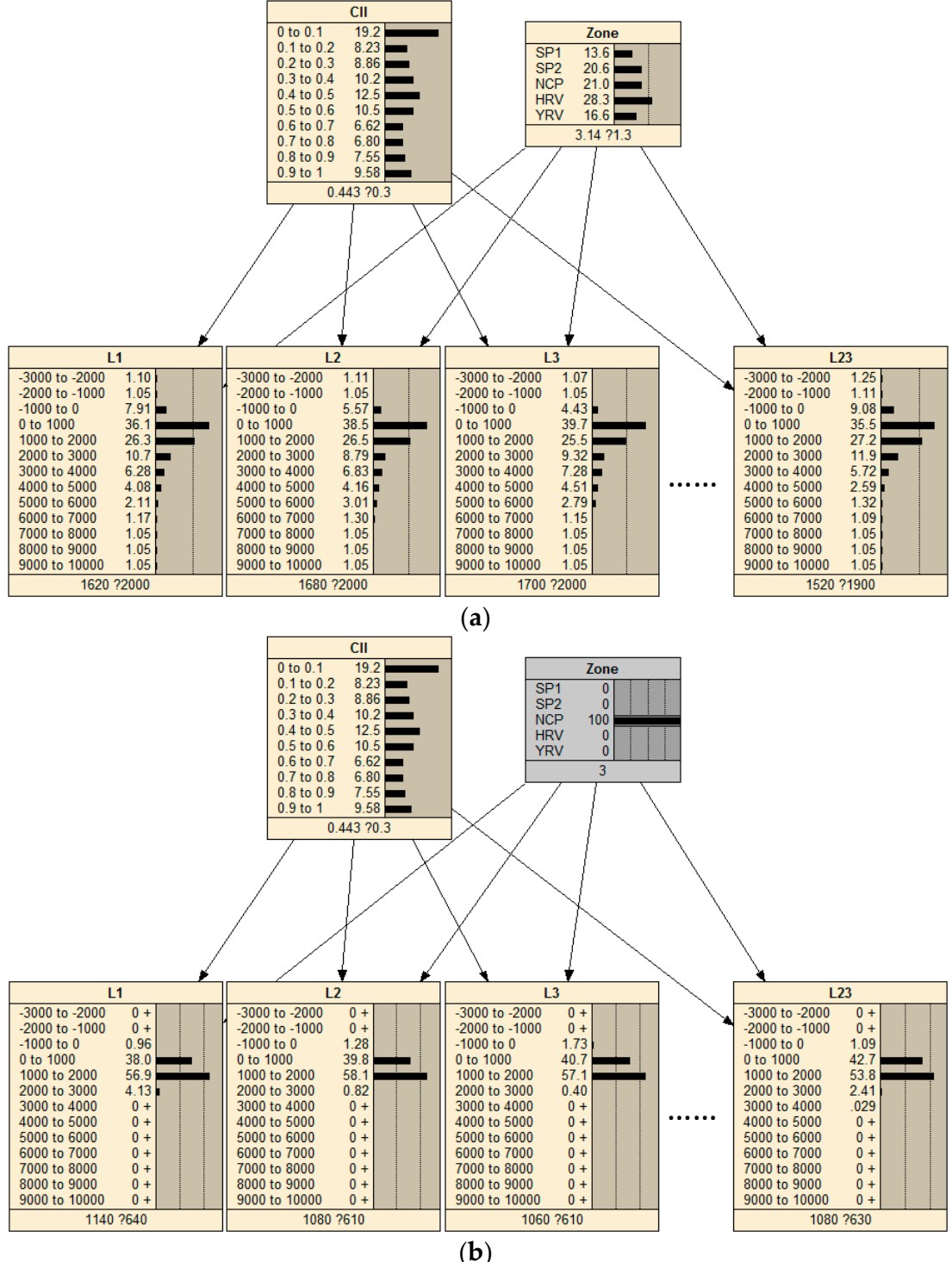

**Figure 5.** The BNPK model (Bayesian network fusing zone data as prior knowledge). SP1, SP2, NCP, HRV, and YRV represent the five sample areas. L1, L2, etc., represent the EVI time-series. (**a**) CPT when adding a zone node as prior knowledge. We simply assigned codes 1–5 to the five sample areas. After adding this node, all the CPTs in other nodes updated automatically. The intra-class variations on CII could be observed, which were consistent with the fact of regional differentiation. (**b**) CPT when giving evidence of zone information. If we provide evidence of zone type as NCP at 100% probability, all the child nodes update the CPT through the propagation of probabilities. This CPT is locally trained and exclusively for NCP.

*3.4. Validation of the Model*

The model was calibrated, and the accuracy of the modeled CII was validated against the sample data (derived from Landsat images).

In the model calibration, the performance of the BN models with or without prior knowledge (named BN and BNPK, respectively) was compared. To test the prediction success rate of the model, using the sample CII values that served as input to the Bayesian network model as reference values, the logarithmic loss, quadratic loss and spherical payoff measures [34] were calculated to compare the prediction abilities of the proposed BN model. The logarithmic loss is a cross entropy estimate with scores between 0 and 1, and values closer to 0 indicate a lower penalty. The quadratic loss is similar to the logarithmic loss score but varies in the interval [0, 2], with 0 being the best. The spherical payoff varies in the interval [0, 1], with 1 representing the best classifier performance [35]. These results are calculated in the standard way for scoring rules, and their respective equations are:

$$\text{Logarithmic loss} = M(-\ln S) \tag{7}$$

$$\text{Quadratic loss} = M\left(1 - 2S + \sum_{j=1}^{n} P_j^2\right) \tag{8}$$

$$\text{Spherical payoff} = M\left(\frac{S}{\sum_{j=1}^{n} P_j^2}\right) \tag{9}$$

where $M$ is the mean probability value of a given state averaged over all cases; $S$ is the probability predicted for the correct state of class; $P_j$ is the probability predicted for class $j$; and $n$ is the number of states for which the training data provides a value for the classification variable.

When validating the accuracy of the modeled CII, both the modeled result and the sample data from the Landsat 8 images were aggregated to 2000 m fractural images by averaging the values within each block to facilitate analysis. The accuracy of the model outputs was tested using the coefficient of determination $R^2$, root mean square error (RMSE), intercept $a$ and slope $b$ of the simple linear regression with respect to the validation samples. After the validation tests, the BN model was complete and could then be applied to the mapping of CII in the study area.

## 4. Results and Analysis

*4.1. Model Calibration*

We used 30% of the sample data as training data and the rest as validation data. The accuracy of the model outputs was tested using the relative error rate with respect to the validation samples. The relative error rates were 47.75% and 27.68% for the two models, respectively (Table 3).

**Table 3.** Overall accuracy measures of the models.

|      | Relative Error (%) | Logarithmic Loss | Quadratic Loss | Spherical Payoff |
|------|--------------------|------------------|----------------|------------------|
| **BN**   | 47.75 | 0.46 | 0.51 | 0.72 |
| **BNPK** | 27.68 | 0.17 | 0.22 | 0.93 |

The scatter diagrams of the result predicted by the two models versus the sample data are plotted in Figure 6. The test samples in the five areas were put together to draw these figures. We used the method of least absolute residuals (LAR) when fitting the data. The LAR method finds a curve that minimizes the absolute difference of the residuals, rather than the squared differences. The fitting output with the R$^2$ of 0.44 and 0.79 and the *p*-value of 0.59 and 0.14 for the two models was obtained at the 95% confidence level. The BN model had poor prediction because it ignored the intra-class

variations in the MODIS time-series. The BNPK model predicted CII better than BN with a $R^2$ of 0.79 because of its full consideration of prior knowledge.

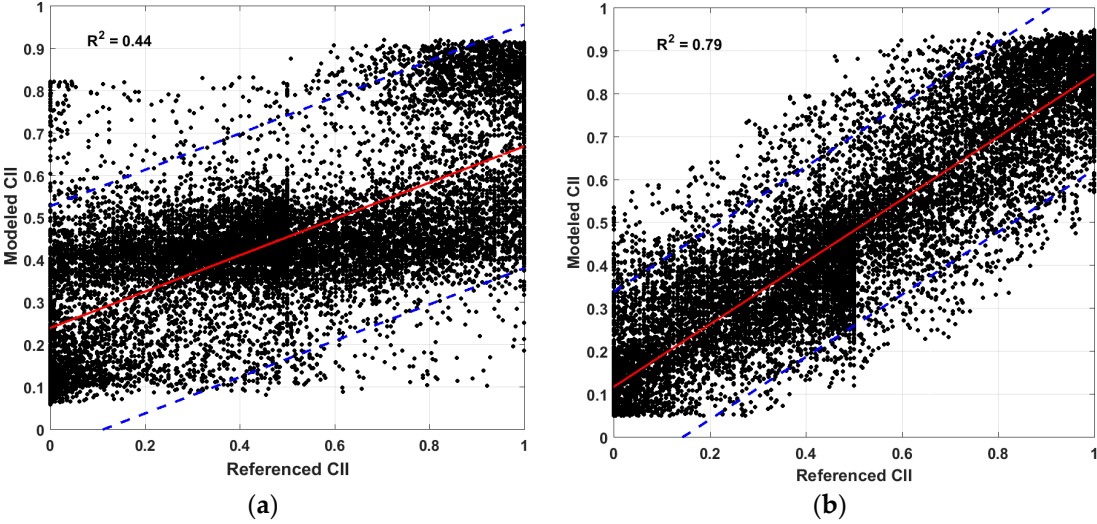

**Figure 6.** Trend line of the modeled and the referenced CII at pixel level, (**a**) BN model, and (**b**) BNPK model.

The coefficients of the simple linear regression for the five sample areas are given in Table 4. The BNPK model clearly provides more accurate estimate than the traditional BN model. At pixel level, the highest $R^2$ for the BNPK model approached 0.84, while the BN reached a $R^2$ maxima of 0.679 in HRV. Overall, the BNPK based estimates were better than that of BN in all of the sample areas.

**Table 4.** Coefficient of determination $R^2$, intercept **a** and slope **b** of the simple linear regression for test samples at the pixel level.

|  | BN | | | | BNPK | | | |
|---|---|---|---|---|---|---|---|---|
|  | $R^2$ | RMSE | a | b | $R^2$ | RMSE | a | b |
| **SP1** | 0.424 | 0.081 | 0.280 | 0.405 | 0.501 | 0.094 | 0.144 | 0.553 |
| **SP2** | 0.269 | 0.131 | 0.300 | 0.455 | 0.484 | 0.136 | 0.180 | 0.589 |
| **NCP** | 0.440 | 0.148 | 0.159 | 0.495 | 0.541 | 0.154 | 0.168 | 0.623 |
| **HRV** | 0.679 | 0.147 | 0.139 | 0.655 | 0.836 | 0.117 | 0.089 | 0.810 |
| **YRV** | 0.450 | 0.151 | 0.137 | 0.506 | 0.620 | 0.138 | 0.152 | 0.670 |

*4.2. Accuracy Validation*

First, we compared the modeled CII with the sample CII through visual interpretation. Five patches from the sample areas were selected to draw these figures (Figure 7). We found that the CII samples were well consistent with the modeled result and showed the detailed cropping frequency information.

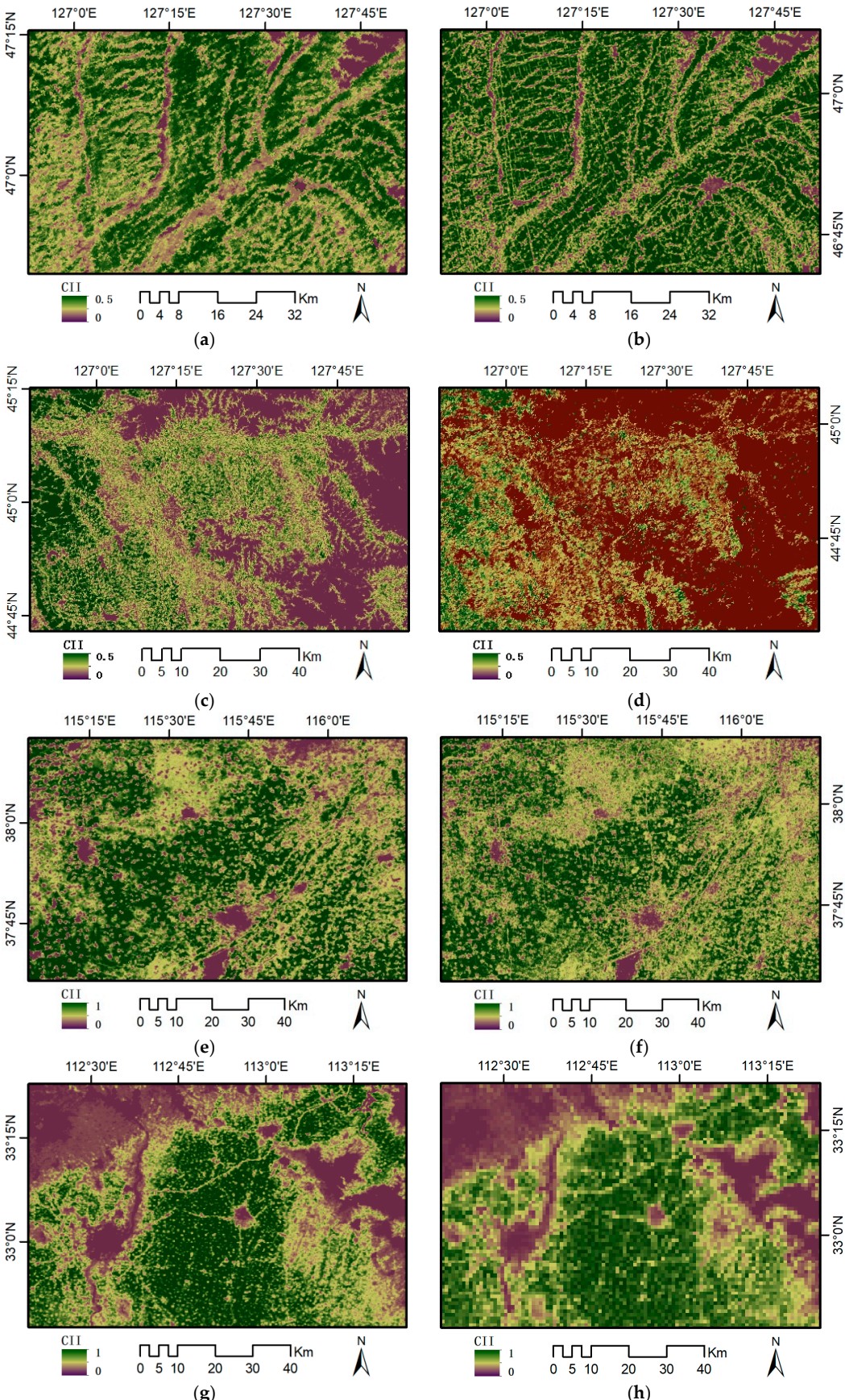

**Figure 7.** *Cont.*

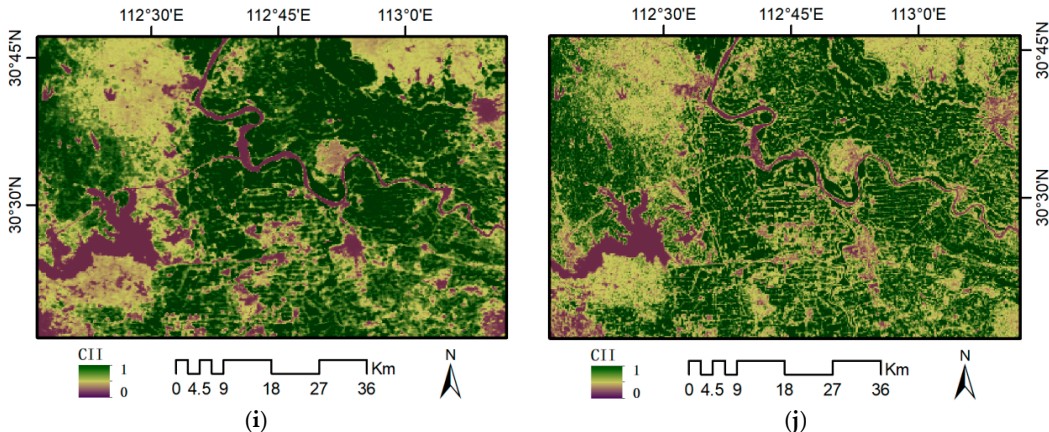

**Figure 7.** Comparison of the modeled and sample CII. (**a**) and (**b**) SP1; (**c**) and (**d**) SP2; (**e**) and (**f**) NCP; (**g**) and (**h**) HRV; and (**i**) and (**j**) YRV.

The modeled CII were analyzed and compared with the referenced data at a 2 km block level (Figure 8). The sample data from Landsat were used as the reference to evaluate the accuracies of the modeled CII. The maximum CII value for SP1 and SP2 was 0.5 because single cropping was the only crop pattern on the Songliao Plain.

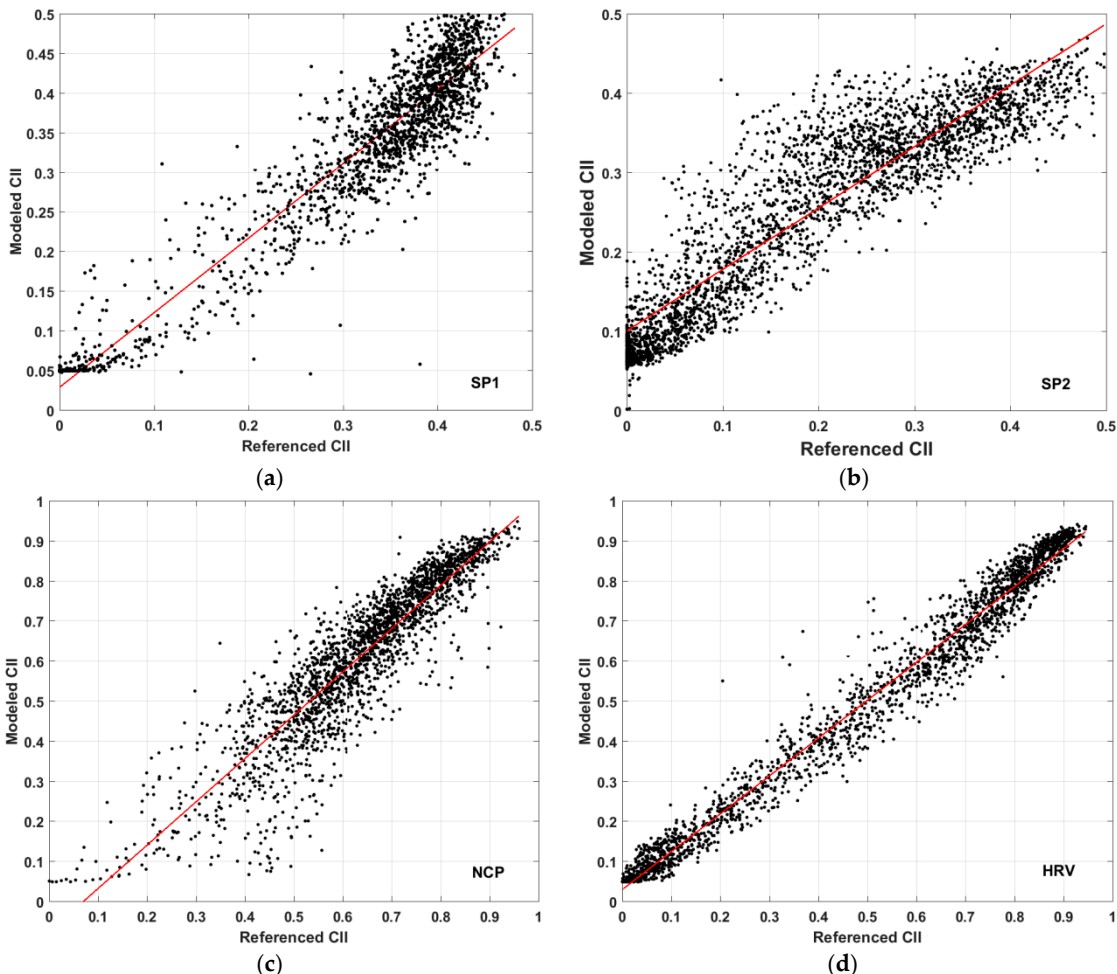

**Figure 8.** *Cont.*

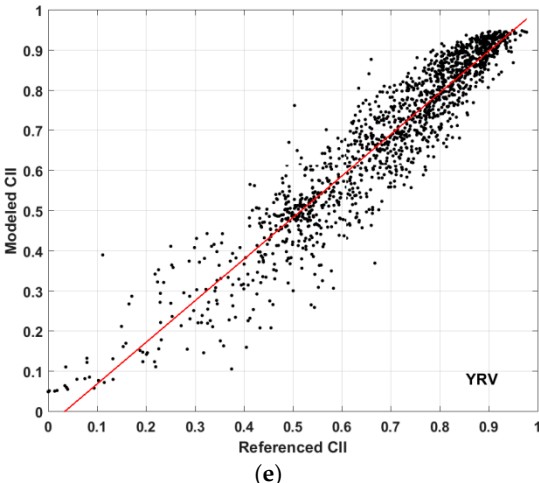

**Figure 8.** Trend line of the modeled and the referenced data at block level. (**a**) SP1, (**b**) SP2, (**c**) NCP, (**d**) HRV, and (**e**) YRV.

The initial pixel-based accuracies increased drastically when the 250 m estimates were aggregated to 2 km levels. At a pixel level, the BNPK model gave acceptable results. However, if the pixel-estimates averaged over more relevant spatial units, then the precision increased essentially and the bias decreased.

The coefficients of the simple linear regression between the referenced and the modeled CII at the block level are given in Table 5. For unbiased estimation, the regression line should approach the 1:1 diagonal line (intercept $a \approx 0$ and slope $b \approx 1$). Slope $b$ was close to 1 and intercept $a$ was relatively small, demonstrating the good fit of the regression.

**Table 5.** Coefficient of determination $R^2$, RMSE, intercept $a$ and slope $b$, $p$-value of the simple linear regression between the referenced, and the modeled CII at the 95% confidence level at the block level.

|  | $R^2$ | RMSE | $a$ | $b$ | $p$-Value |
|---|---|---|---|---|---|
| **SP1** | 0.87 | 0.044 | 0.029 | 0.941 | 0 |
| **SP2** | 0.823 | 0.049 | 0.051 | 0.973 | 0.04 |
| **NCP** | 0.82 | 0.082 | −0.074 | 1.08 | 0 |
| **HRV** | 0.97 | 0.046 | 0.030 | 0.945 | 0 |
| **YRV** | 0.89 | 0.067 | −0.034 | 1.036 | 0 |

## 5. Discussion

Machine-learning-based interpretation of remotely sensed data typically involves models and algorithms that can combine evidence from what is being sensed (for example, the MODIS time-series in this research) with prior knowledge. Mapping cropping intensity index (CII) from MODIS time-series on a large-scale is an example of such a task. Most existing studies have been carried out either on a small-scale or a large-scale, but without considering the regional differentiation factors [27,36–40]. The experiments clearly demonstrated the regional variations within the EVI time-series including the EVI amplitude and phenological stage of the crops (Figure 9). The existing studies have dealt with regional differentiation either through using phenological information to modify the EVI profiles [41,42], or by dividing the study area into independent zones and then mapping each zone separately [43,44].

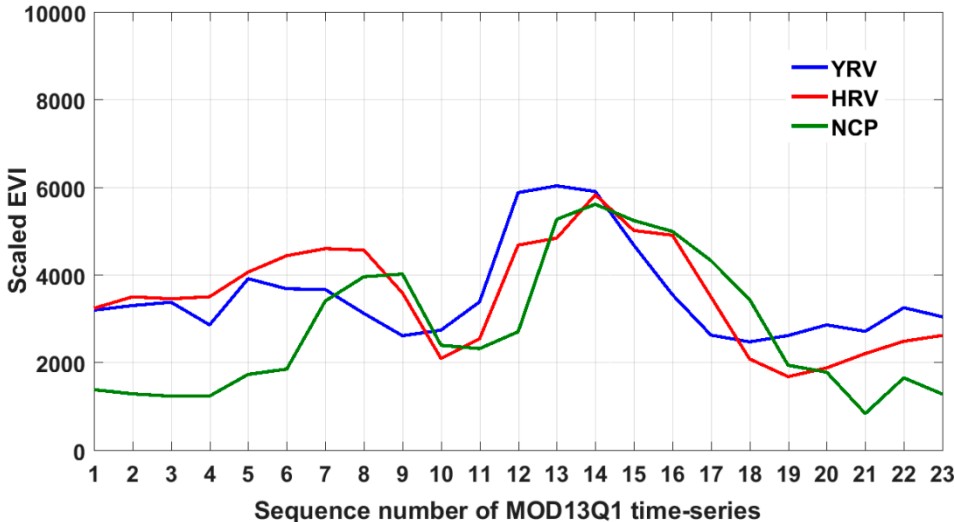

**Figure 9.** EVI profiles of the cropland pixels in three sample areas. About 100 representative pixels with a CII value of 0.7 were selected randomly for each sample area (using NCP, HRV, and YRV as examples), and the mean values of the EVI time-series were calculated to plot this figure.

The BNPK model (Bayesian network model fusing prior knowledge) provides a unique way of combining the MODIS time-series and regional differentiation factors and can be applied to large-scale CII mapping. The BNPK model was constructed with the simplest network structure while still maintaining the ability to model CII with an acceptable error rate. The model was very simple because the only modification to a conventional BN model was to add a node into the model to label where the case came from. The model was then locally trained and at the same time globally optimal. Bayesian networks provide a useful way of dealing with such problems because they combine the robustness of probabilistic methods with the expressiveness of graphs that encode relationships between variables, offering a framework for handling uncertainty and complexity in the estimation of CII within a single model.

The proposed CII is a new indicator which can be use to map cropping intensity at a finer scale compared with the existing researches [10,12–14]. The modeled CII is a fractional image at the sub-pixel level, which is a kind of interval measurement. CII map provides detailed spatial information of cropping intensity, where the spatial heterogeneity of cropping intensity within a MODIS pixel can be accounted for. Compared with cropping frequency, which is a kind of ordinal measurement, CII has numeric scales where we know not only the order, but also the exact differences between the cropping intensity levels. Therefore high spatial and temporal resolution CII datasets can meet the requirements of fine cropping intensity mapping. The increments of cropping intensity are known, consistent, and measurable.

The predicted CII map had high accuracies considering the study area spanned a vast territory. The pixel-based results were less than ideal, however the 2 km block aggregation results were satisfactory. Considering that the resolution of 2 km is sufficient in large-scale estimation, the CII was superior to cropping frequency and MCI. Our research presented a framework to obtain a CII map with both a finer spatial resolution and a fine temporal resolution at a national scale (label C in Figure 10). The conventional methods only provided cropping intensity data with either coarse spatial resolution (label A), or coarse temporal resolution (label B).

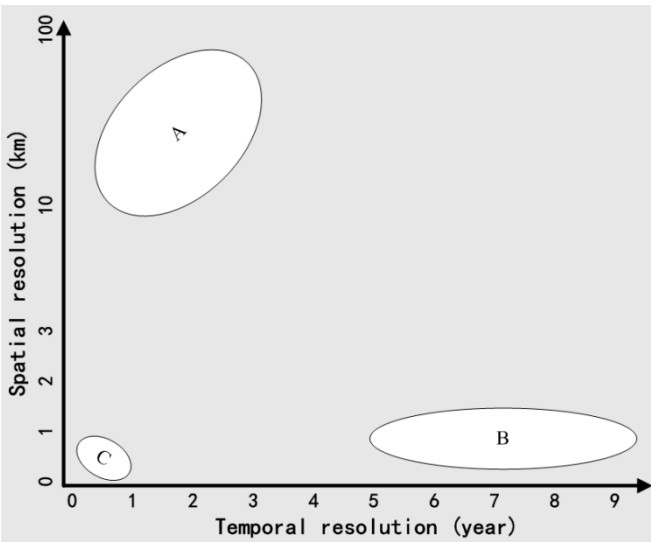

**Figure 10.** Temporal and spatial scale of cropping intensity.

The BNPK model can be applied to any large-scale land surface parameter mapping, provided zonal layer data are available. The delineation of zonal data is then a key issue for extending the applications of the model. For precise mapping, using latitude data together with zonal data instead of only zonal data requires more attention in the future. This model is also applicable for any given year with MODIS data coverage, provided that cropping frequency sample data from high-resolution cloud-free data are available.

## 6. Conclusions

The cropping intensity index (CII) is not only the basic data for agricultural remote sensing but also an important input variable for ecosystem modeling and global change research. This paper reports our work on estimating CII in mainland China from time-series MODIS data. A CII was designed and a BNPK model (Bayesian network model fusing prior knowledge) fusing regional differentiation factor as prior knowledge was built and applied to the estimation of CII. The novelty and contributions of our work are summarized as follows. First, the proposed model had the superiority of fusing prior knowledge into the BN model, greatly decreasing the uncertainty in large-scale CII mapping. The model has the advantages of considering both the EVI profiles and regional differentiation when estimating the CII over large-scale areas. Second, the proposed CII is an interval measurement and can map cropping intensity with both a finer spatial resolution and a fine temporal resolution at a national scale. Conventional approaches for national scale crop mapping have certain difficulties in their application. The proposed CII and BNPK model can therefore be a good solution to obtain a more accurate cropping intensity map.

The pixel-based results were not satisfactory, partly because of the data quality and the coarse resolution of the MODIS data. Diverse crop patterns within even one sample area can also lead to uncertainty of the model. Future work will involve more precise delineation of zonal data according to crop patterns and the analysis of spatial and temporal dynamics of CII.

**Author Contributions:** Conceptualization, J.T. and W.W.; methodology, J.T.; validation, J.T. and M.X.; formal analysis, J.T.; investigation, M.X.; resources, J.T.; data curation, M.X.; writing—original draft preparation, J.T.; writing—review and editing, J.T. and W.W.; supervision, J.T.; project administration, J.T.; funding acquisition, W.W.

**Funding:** This research was funded by the National Natural Science Foundation of China (grant number 41871356), and the Natural Science Foundation of Hubei Province (grant number 2017CFB434).

**Acknowledgments:** The authors would like to thank the anonymous reviewers for their comments and suggestions regarding this paper.

**Conflicts of Interest:** The authors declare no conflict of interest.

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
