# Peer review of "Using the Bayesian Network to Map Large-Scale Cropping Intensity by Fusing Multi-Source Data"

_remotesensing, doi:10.3390/rs11020168_

Reviewer 1 Report

The manuscript is dedicated to very important domain - agricultural activities estimation and crop monitoring. But authors should take into account:

Due to different datasources usage the manuscipt should have another title.

The paper should have strong definition of "crop intensity" term. Authors have annalysed and mentioned different senses of that term, but in the paper they use another semantic. Besides that, formula (1) and it's sense should be described in more detail.

It's unclear, how in proposed procedure authors propose to use absolutely different from year to year class spatial distributions due to crop rotaion requirements and weather conditions?

How to take into account different vegetation seasons and provide the opportunities to use it for any other year?

Author Response

We have made substantial revision on this version, including adjusting the structure of the paper, modifying the definition of CII, removing tedious experimental process descriptions, adding some statistical validation of results. Because there so many changes in the new version, for brevity some modifications do not use the tracked mode.

Comments and Suggestions for Authors

Reviewer 1

The manuscript is dedicated to very important domain - agricultural activities estimation and crop monitoring. But authors should take into account:

1.Due to different datasources usage the manuscipt should have another title.

Answer: We have changed the title, replaced ‘based on MODIS time-series’ by ‘fusing multi-source data’.

2. The paper should have strong definition of "crop intensity" term. Authors have annalysed and mentioned different senses of that term, but in the paper they use another semantic. Besides that, formula (1) and it's sense should be described in more detail.

Answer: We have made substantial revision on this part, modifying the definition of CII based on Bayesian inference. The inference of Bayesian Network was moved to the front of CII definition, to make this part more reasonable and clearer.

3. It's unclear, how in proposed procedure authors propose to use absolutely different from year to year class spatial distributions due to crop rotaion requirements and weather conditions?

How to take into account different vegetation seasons and provide the opportunities to use it for any other year?

Answer: The experiment was conducted on 2016 data only. But it is repeatable in any year given MODIS and high-resolution sample data. Some related information and discussion have been added to the starting part of Section 2.2 and Discussion Section.

Reviewer 2 Report

Dear Authors,

Thank you for submitting your work to Remote Sensing. My general impression of your work is good but some questions are in place regarding the following: 1) the proposed index (CI) is useful only in monoculture and two-culture sown agricultural environments. According to your claims the studied regions fall completely in those two types or categories in respect to the pixel size of MODIS (EVI) products. To my view, however, the proposed index is not universal due to its shortages to address all the crop variability which exist in various environments as well as in very diverse and small crop fields. This shortage has to be addressed in your methodology. 2) The PCA is applied to a multi-temporal EVI series which in effect decreases the dimensionality in time domain. What is the main idea of not making use of the time and how much of the variability (%) the three PC account for? 3) the used Zones in the NB classification are based on administrative regions or provinces, would it be better to account for agro-climatic zoning instead which is more valuable in terms of crops?

Last but not least, please consult the attached manuscript for my minor comments and suggestions.

Kind regards,

Reviewer

Author Response

We have made substantial revision on this version, including adjusting the structure of the paper, modifying the definition of CII, removing tedious experimental process descriptions, adding some statistical validation of results. Because there so many changes in the new version, for brevity some modifications do not use the tracked mode.

Comments and Suggestions for Authors

Reviewer 2

Thank you for submitting your work to Remote Sensing. My general impression of your work is good but some questions are in place regarding the following:

1) the proposed index (CI) is useful only in monoculture and two-culture sown agricultural environments. According to your claims the studied regions fall completely in those two types or categories in respect to the pixel size of MODIS (EVI) products. To my view, however, the proposed index is not universal due to its shortages to address all the crop variability which exist in various environments as well as in very diverse and small crop fields. This shortage has to be addressed in your methodology.

Answer: Firstly, there are no tri-season crops in the research area (this information has been added to Sub-section 2.1). The descriptions of ‘…within a MODIS pixel’ is talking about preparing the sample data, no longer about definition of CII. This revision makes this part more clear.

      The intensity and variability of EVI within a natural year can reflect the fragmentation of fields and areal coverage of given crop pattern. This is the basis of our definition of CII. The new definition of CII in Sub-section 3.2 and a flowchart in Sub-section 3.3 will make it clearer.

2) The PCA is applied to a multi-temporal EVI series which in effect decreases the dimensionality in time domain. What is the main idea of not making use of the time and how much of the variability (%) the three PC account for?

Answer: We found that three PCA components account for only about 70% of the variability, so we changed the features by using EVI time-series. PCAs can simplify the BN network structure and reduce the computational complexity, but at the same time will result in information loss. We have made substantial revision on this part.

3) the used Zones in the NB classification are based on administrative regions or provinces, would it be better to account for agro-climatic zoning instead which is more valuable in terms of crops?

Answer: Yes, thank you for your good suggestion. We have changed the administrative regions to agro-climatic zones to make it more reasonable.

Last but not least, please consult the attached manuscript for my minor comments and suggestions.

Answer: We have adopted almost all the minor comments and revised the manuscript. But we have a few expressions remain unchanged, along with some explanations.

(a) monoculture? We still use ‘single cropping’, since in the manuscript we use ‘single cropping’, ‘double cropping’ and ‘triple cropping’ to describe different crop pattern. (b) Do you have high resolution Chinese EO data over these study areas? Yes we have. But Landsat 8 data can meet our need in this research. (c)Doesn't this limit the applicability of the proposed index to the chosen test areas only? When talking about crop pattern, existing researches often deal with single cropping and double cropping only. Actually there are little triple cropping in China now (almost 1~3% of the total cropland). Another reason is that the existing few triple cropping croplands locate in the fragmented South China and have small field size. Hope this explanation makes sense. If you have more suggestions please feel free to let me know.

Thank you for your good suggestions on improving the quality of the manuscript.

Reviewer 3 Report

The paper illustrated the mapping of large-scale cropping intensity based on MODIS time-series by using Bayesian Network in mainland China. The paper needs a critical modification for publication. Questions and important modifications are given below:

Comments

a)     Introduction part is not sufficient in this study. More recent papers should be reviewed. It should be more interesting.

b)     Results are not sufficient for this study. The more statistical analysis should be required. Please provide p-value for all scatter diagrams.

c)     What is the duration (length) of the data (MODIS/Landsat) of this study and also mention the all available datasets in a year? Can you show the single crop/ double crop maps and land use map?

d)     Results should be more analyzed in the revised manuscript.

e)     The paper is looking a software exercise. The methodology should be more clear.

f)      Author can be improved the writhing English language.

Author Response

We have made substantial revision on this version, including adjusting the structure of the paper, modifying the definition of CII, removing tedious experimental process descriptions, adding some statistical validation of results. Because there so many changes in the new version, for brevity some modifications do not use the tracked mode.

Comments and Suggestions for Authors

Reviewer 3

The paper illustrated the mapping of large-scale cropping intensity based on MODIS time-series by using Bayesian Network in mainland China. The paper needs a critical modification for publication. Questions and important modifications are given below:

Comments

a)     Introduction part is not sufficient in this study. More recent papers should be reviewed. It should be more interesting.

Answer: A few latest literatures (2017 and 2018) have been added to the manuscript.

b)     Results are not sufficient for this study. The more statistical analysis should be required. Please provide p-value for all scatter diagrams.

Answer: We provided the p-value and confidence level for all the scatter figures. We also gave the relative error rate of the model to make the calibration more believable.

c)     What is the duration (length) of the data (MODIS/Landsat) of this study and also mention the all available datasets in a year? Can you show the single crop/ double crop maps and land use map?

Answer: The information about the duration of the data has been added to Sub-section 2.2. We have modified Figure 1 by adding the single cropping / double cropping distribution information to the figure. As for the land-cover map, since we focused on cropland so we will not include other land-cover type in the figure. Hope this explanation makes sense. Further comments are appreciated.

d)     Results should be more analyzed in the revised manuscript.

Answer: We have added the comparison of the modeled CII with sample CII through visual interpretation to the manuscript.

e)     The paper is looking a software exercise. The methodology should be more clear.

Answer: We have revised and shortened the Sub-section 3.3 and 4.1 to make it more precise and clearer.

f)      Author can be improved the writhing English language.

Answer: The language has been partly improved. If it does not meet the standard of publication, we will apply for a language editing service. Thank you.

Round  2

Reviewer 1 Report

The comments are still actual. Probability is very good tool. But for applied problems it should be illustrated, how we can use it and link to real data.

Landsat data are mentioned as reference data only, without any definite description of this part of main workflow.

Author Response

The comments are still actual. Probability is very good tool. But for applied problems it should be illustrated, how we can use it and link to real data.

Answer: We revised Sub-section 3.3.1 by adding a figure (Figure 4), a table (Table 2), a formula (formula 6), and a few sentences. These revisions together with Sub-section 3.2 will make it clearer, illustrating how to link CPT to real data.

Landsat data are mentioned as reference data only, without any definite description of this part of main workflow.

Answer: Actually we have introduced the Landsat data covered the sample areas (Sub-section 2.2), and the method to derive cropping frequency samples from Landsat data (Sub-section 3.2). If you have any more comments, feel free to advice. Thank you.

Reviewer 2 Report

Dear Authors,

Thank you very much for the detailed reply and the revised version of your manuscript. I think that now your work has the maturity to being published. I have one more suggestion before that, however, and it is to use PROBA-V 100 m S10 data. PROBA_V does not have an EVI index since it has no blue band for its derivation but it has twice as better spatial resolution compared to MODIS. However, it also depends on your temporal window. Nevertheless, you can consider in future PROBA-V, MERIS and Sentinel-3 satellite products (also available for free) to add to your analysis.

Kind regards,

Reviewer 2

Author Response

Thank you very much for the detailed reply and the revised version of your manuscript. I think that now your work has the maturity to being published. I have one more suggestion before that, however, and it is to use PROBA-V 100 m S10 data. PROBA_V does not have an EVI index since it has no blue band for its derivation but it has twice as better spatial resolution compared to MODIS. However, it also depends on your temporal window. Nevertheless, you can consider in future PROBA-V, MERIS and Sentinel-3 satellite products (also available for free) to add to your analysis.

Answer: Thank you very much for your good suggestions.

Reviewer 3 Report

Accepted

Author Response

Accepted

Answer: Thank you very much for your good suggestions.